# Pharmaceutical Assessment Suggests Locomotion Hyperactivity in Zebrafish Triggered by Arecoline Might Be Associated with Multiple Muscarinic Acetylcholine Receptors Activation

**DOI:** 10.3390/toxins13040259

**Published:** 2021-04-03

**Authors:** Petrus Siregar, Gilbert Audira, Ling-Yi Feng, Jia-Hau Lee, Fiorency Santoso, Wen-Hao Yu, Yu-Heng Lai, Jih-Heng Li, Ying-Ting Lin, Jung-Ren Chen, Chung-Der Hsiao

**Affiliations:** 1Department of Chemistry, Chung Yuan Christian University, Chung-Li, Taoyuan City 320314, Taiwan; siregar.petrus27@gmail.com (P.S.); gilbertaudira@yahoo.com (G.A.); 2Department of Bioscience Technology, Chung Yuan Christian University, Chung-Li, Taoyuan City 3020314, Taiwan; fiorency_santoso@yahoo.co.id; 3School of Pharmacy and Ph.D. Program in Toxicology, Kaohsiung Medical University, Kaohsiung 80708, Taiwan; joanna@kmu.edu.tw; 4Substance and Behavior Addiction Research Center, Kaohsiung Medical University, Kaohsiung 80708, Taiwan; 5Department of Biotechnology, College of Life Science, Kaohsiung Medical University, Kaohsiung 80708, Taiwan; u105831002@kmu.edu.tw (J.-H.L.); u104022019@kmu.edu.tw (W.-H.Y.); 6Graduate Institute of Natural Products, College of Pharmacy, Kaohsiung Medical University, Kaohsiung 80708, Taiwan; 7Department of Chemistry, Chinese Culture University, Taipei 11114, Taiwan; yh21@ulive.pccu.edu.tw; 8Drug Development & Value Creation Research Center, Kaohsiung Medical University, Kaohsiung 80708, Taiwan; 9Department of Biological Science & Technology, College of Medicine, I-Shou University, Kaohsiung 82445, Taiwan; jrchen@isu.edu.tw

**Keywords:** arecoline, locomotion, zebrafish, betel nut, muscarinic acetylcholine receptor, molecular docking, antagonist

## Abstract

Arecoline is one of the nicotinic acid-based alkaloids, which is found in the betel nut. In addition to its function as a muscarinic agonist, arecoline exhibits several adverse effects, such as inducing growth retardation and causing developmental defects in animal embryos, including zebrafish, chicken, and mice. In this study, we aimed to study the potential adverse effects of waterborne arecoline exposure on zebrafish larvae locomotor activity and investigate the possible mechanism of the arecoline effects in zebrafish behavior. The zebrafish behavior analysis, together with molecular docking and the antagonist co-exposure experiment using muscarinic acetylcholine receptor antagonists were conducted. Zebrafish larvae aged 96 h post-fertilization (hpf) were exposed to different concentrations (0.001, 0.01, 0.1, and 1 ppm) of arecoline for 30 min and 24 h, respectively, to find out the effect of arecoline in different time exposures. Locomotor activities were measured and quantified at 120 hpf. The results showed that arecoline caused zebrafish larvae locomotor hyperactivities, even at a very low concentration. For the mechanistic study, we conducted a structure-based molecular docking simulation and antagonist co-exposure experiment to explore the potential interactions between arecoline and eight subtypes, namely, M1a, M2a, M2b, M3a, M3b, M4a, M5a, and M5b, of zebrafish endogenous muscarinic acetylcholine receptors (mAChRs). Arecoline was predicted to show a strong binding affinity to most of the subtypes. We also discovered that the locomotion hyperactivity phenotypes triggered by arecoline could be rescued by co-incubating it with M1 to M4 mAChR antagonists. Taken together, by a pharmacological approach, we demonstrated that arecoline functions as a highly potent hyperactivity-stimulating compound in zebrafish that is mediated by multiple muscarinic acetylcholine receptors.

## 1. Introduction

Betel nut (*Areca catechu*) chewing is a popular habit and tradition in Taiwan and other countries in Southeast Asia. Sense of euphoria, heightened alertness, hot sensation in the body, and increased capacity in work have been claimed by those who chew betel nuts [1]. Betel nut chewing becomes popular since they elicit a psychoactive effect in humans [2]. In addition, betel nut is also listed as the most addictive substance in the world along with tobacco, alcohol, and caffeine [3]. The psychoactive effect in betel nut is associated with arecoline (methyl 1,2,5,6-tetrahydro-1-methyl nicotinate) [4], which is the most abundant nicotinic acid-based alkaloid in betel nut [5]. Arecoline is a natural alkaloid that has cyto-modulating effects that correlate with oral cancer [6]. Furthermore, arecoline alkaloid is able to promote peristalsis and glandular secretion by acting on both muscarinic and nicotinic receptors [1,7]. International Agency for Research on Cancer (IARC) has found evidence demonstrating that chewing betel nut could cause oral cancer in humans [3]. However, arecoline can be used for medicinal purposes such as anti-helminthic, mental illness drugs, digestive agents, as well as psychosomatic medicine [1].

To date, arecoline has been extensively studied due to its potential adverse effect on humans or animals. Chick embryos injected with arecoline showed developmental abnormalities, such as reduced body size, scanty feathering, and skeletal defects, suggesting that arecoline acts as a fetotoxic agent causing deleterious and teratogenic effects during development [8]. Another study in the mouse model showed that arecoline resulted in embryotoxicity and fetopathic changes in pregnant mice, including fetal death and a decrease in fetal weight without major morphological deformities [9]. Meanwhile, in zebrafish studies, it has been proved that 0.01–0.04% (100–400 ppm) of arecoline caused general growth retardation and a lower heart rate [10]. Another experiment with zebrafish larvae that used 0.001–0.04% (10–400 ppm) arecoline also generate developmental retardation and morphological deformities [3]. All these results showed that arecoline in a higher concentration (>1 ppm) has a potentially adverse effect on the animal, especially in the developmental stage. Based on these experiments, we design our study to exposed arecoline in several concentrations (0.001–1 ppm) and make sure developmental retardation did not interfere with the locomotor result. These concentrations were much less than the daily dose of arecoline consumed by betel chewers or the dose used for the treatment in Alzheimer’s patients, which is 9.6–61 mg/day [11,12].

Whilst the majority of the research uses developmental toxicity in zebrafish analysis, others have also used a behavioral analysis since it is often more sensitive and informative compared to other toxicological and pharmacological evaluations [13,14,15]. Some research proved the effects of arecoline using animal behavior. It is thought that arecoline exposure often leads to hyperactivity since arecoline can generate the sensation of euphoria [16,17,18]. The hypothesis behind arecoline’s potential ability in inducing hyperactivity comes from the capability of arecoline to stimulate the body into a state of excitement (body excitability) by stimulating the mAChRs. However, different results were shown in adult zebrafish. The acute treatment of arecoline caused an anxiolytic-like behavior that resembles anxious movement and stimulant effects of nicotine in zebrafish [19,20], the behavior indicated by a low level of locomotor activity. Similarly, another study also discovered that higher concentrations (10 ppm) of arecoline caused hypoactivity in adult zebrafish [21]. According to these studies, they showed inconsistent results. This is intriguing, how exactly the arecoline potential adverse effect could influence the locomotor activity. It is also important to seek the possible mechanism behind it. To understand that, we used a behavioral assay to grasp the mechanism behind its adverse effect in larvae zebrafish. Some studies have utilized the advantage of behavioral assays, especially in zebrafish larvae in toxicology studies [22,23,24]. With the advantage of zebrafish and behavioral assay, we hope to uncover the arecoline pathway in affecting locomotor activity.

With a probability of arecoline affecting locomotor activity via the muscarinic acetylcholine receptor, it is necessary to find out the connection between arecoline and mAChRs. Several previous studies have provided clues for a connection between arecoline and mAChR. Calogero et al. showed in rats that arecoline could stimulate the HPA axis through a centrally mediated CRH mechanism via the mAChRs [25]. Furthermore, it also has been shown that arecoline was able to produce antinociception by activating the M1 muscarinic receptor subtype in mice [26]. Taken together, the activation of multiple mAChRs could become a tantalizing possible explanation underlying the arecoline effects observed in animals or humans. Thus, in this study, we wanted to observe the effects of arecoline in the zebrafish larvae locomotor activity. To pursue this idea further, zebrafish were used as an animal model in the current study, since not only are they a good model for behavioral analysis, but also they possess all five mAChRs genes. Previously, it was well established that the vertebrate predecessor went through two rounds of whole-genome duplication [27]. In addition, the predecessor also went through a third tetraploidization (3R) after the divergence [28]. Interestingly, the third tetraploidization (3R) event double or duplicate the repertoire of all five mAChR genes once more, which results in all the 10 genes that were also present in zebrafish [29]. With those advantages, zebrafish is the perfect model for investigating whether the arecoline could bind to the zebrafish mAChR.

A dry lab in silico molecular simulation was conducted to provide more insights into the ligand-receptor interactions at the atomic scale. The in silico structure-based molecular simulation provides a powerful tool to illustrate potential ligand-receptor binding scenarios. Remarkable improvements in computational capacity and efficiency have made molecular docking a useful and accessible tool to show the molecular interactions between the ligand and receptor within the atomic level [30,31]. This approach is especially crucial for drug discovery/development or toxicity predictions. The outcomes of in silico molecular simulation can also guide in browsing hit drugs and designing new leads [32].

Taken together, even though a prior study already found that arecoline seems to have a potential effect on zebrafish larvae, especially in its potential and effectivity in affecting their locomotion activity, the mechanism behind how arecoline affects locomotion is still poorly understood, especially in vivo [3]. Here, we hypothesized that arecoline could cause hyperactivity by binding and activating multiple mAChRs. Finally, along with the in silico and locomotor activity assay, an antagonist co-exposure experiment whereby several mAChR antagonists were administered together with arecoline was performed to elucidate the potential mechanisms. Using the advantage of behavioral assay together with molecular docking and antagonist experiment, we successfully proved first evidence to support arecoline could bind with multiple mAChRs to induce hyperactivity in larvae zebrafish.

## 2. Results

### 2.1. Low Dose Arecoline Treatment Elevates Locomotor Activity in Zebrafish Larvae

To observe the potential effects of arecoline on locomotor activity in zebrafish larvae, we monitored and compared the swimming activity between untreated control and arecoline-treated zebrafish larvae at 120 hpf (with 24-h arecoline incubation). Overall, a dramatic increase in locomotor activity was observed in treated groups with different doses of arecoline and showed significantly long total distance travelling during the test (Figure 1A). Moreover, the hyperactivity was displayed in treated groups during both light and dark cycles. Larvae zebrafish were naturally more active in the dark cycle [33], as this is one of the characteristics of larvae zebrafish. Even in the dark cycle, arecoline-treated larvae showed a significantly higher locomotor activity compared to the control fish (Figure 1C). In addition, we also found that the distance travelled during the light cycle after exposure to different doses of arecoline showed an increment in all testing doses (Figure 1B,C). To confirm this hyperactivity-like behavior, ethanol was used as a positive control drug since the hyperactivity-like behavior effect of acute exposure to ethanol is well documented [34]. As expected, acute ethanol exposure on zebrafish caused hyperactivity-like behavior in the zebrafish larvae (Figure A4A in Appendix A). The ethanol-treated fish exhibited a higher locomotor activity during the dark cycle while in the light cycle, they displayed a lower locomotor activity compared to the arecoline-treated fish at the same concentration (Figure A4B,C). Taken together, arecoline, even at a low concentration, can induce locomotor activity in zebrafish larvae. One interesting result showed that arecoline has a dose-response curve, as seen in Figure 1C.

To further analyze the hyperactivity behavior caused by the arecoline treatment, the average burst movement in every minute was counted and compared. All of the treated zebrafish larvae displayed significantly higher activities in both light and dark cycles compared to the control group (Figure A2A). This phenomenon was corroborated by higher burst movement counts detected in both cycles. Interestingly, we noticed increased hyperactivity in most of the time points over the light period in a dose-dependent manner (Figure A2B in Appendix A). However, we found decreased burst movement counts together with increased arecoline concentrations during the dark cycle (Figure A2C). To observe the arecoline effect on the movement orientation in fish larvae, the average rotation was analyzed. We found that arecoline affected the movement orientation during both light and dark cycles (Figure A3A). Under the light condition, except for the lowest concentration (0.001 ppm) of the arecoline-treated group, did not show an increased rotation movement while the total rotation counts of the rest of the groups increased (Figure A3B). On the other hand, under the dark condition, all of the arecoline-treated groups displayed a higher total rotation count per minute compared to the control group (Figure A3C). In addition, supporting the Larvae Photomotor Response (LPMR) results, the non-linear dose-response of arecoline was also shown in burst movement (Figure A2B) and average rotation (Figure A2B) count endpoints with higher values of these endpoints in low concentration groups and lower values in high concentration groups.

In order to study the arecoline effects for different incubation times, we compared the induced hyperactivity in zebrafish larvae after 24-h exposure and short-term exposure (30 min). Results showed that the short incubation of arecoline caused a significantly higher locomotion activity than the 24-h incubation group during the light cycle (Figure 2A). However, during the dark cycle, while the distance travelled of the rest of the groups was shorter than that of the 24-h exposure group, the lowest concentration (0.001 ppm) of the arecoline-treated group showed a longer distance travelled than the other groups (Figure 2B). Taken together, the 30 min short-term incubation of arecoline showed different patterns in altering the zebrafish larvae locomotor activity as compared to the effects that resulted from the 24-h incubation. We chose the 24-h exposure since even in the dark cycle where naturally larvae zebrafish are always active, the arecoline treated larvae is able to increase it more compared to the 30-min exposure. This result showed that chronic exposure resulted in a robust outcome.

### 2.2. Low Dose Arecoline Treatment Altered Larvae Photomotor Response (LPMR) in Zebrafish Larvae

To examine zebrafish larvae behavioral responses to a sudden change in lighting condition after the arecoline treatment, LPMR was measured following methods described previously [35,36,37,38]. The LPMR identification used a method different from what was in a previously performed study in which 24 hpf embryos were studied and it was thought to be a non-visual zebrafish larval response to a light stimulus [39]. Similar to the locomotor activity results, the highest concentration of arecoline altered LPMR in zebrafish larvae (Figure 3). This phenomenon was shown by a significant decrement of LPMR in the dark cycle as displayed in the highest concentration (1 ppm) treated group, which was exactly opposite from the other concentrations. Furthermore, based on this result, there is a possibility that arecoline might have a U-shaped dose-response curve since in the LPMR test, especially in the dark cycle, the intermediate concentration of arecoline caused the most pronounced effects compared to other concentrations, including the highest concentration. However, the rest of the treated groups displayed only slight increments of LPMRs in both light and dark conditions.

### 2.3. Molecular Docking for Arecoline and Muscarinic Acetylcholine Receptor

Briefly, after homology modeling structures of zebrafish muscarinic acetylcholine receptor (chrm) subtypes were made. The amino acid sequence (aa) of zebrafish (*Danio rerio*) chrm subtypes were selected from the Uniprot database and their corresponding IDs were listed. The template structures for modeling structures were picked by NCBI blastp with the available protein database (PDB) crystal protein structures (with 74.05% to 90.29% protein sequence identities). All homology modeling structures for molecular docking were constructed by the Modeller software. Dock scores were assigned by the molecular docking module, LigandFit, indicating the binding affinity between each target protein and arecoline. Results from molecular docking showed that arecoline may bind to six sub-types of mAChR with dock scores of 35.909 (for M1a), 36.896 (for M2a), 34.155 (for M2b), 38.108 (for M3a), 38.419 (for M3b), 34.285 (for M4a). M5 mAChR, on the contrary, displays a relatively lower dock score of 26.621 for M5a and 1.839 for M5b (Table 1). Even though the docking score should not be used to automatically select the strong binders, it is helpful in deciding the “bad” binders’ elimination. Among all mAChR, arecoline had the highest dock score with M3a and M3b subtypes. Coincidently, M3b also has the highest dock score among the B subtypes. Furthermore, the detailed interactions between arecoline and M3a were investigated (Figure 4A). We identified putative binding pockets in the middle of the subtype homology 3D models. Arecoline formed hydrophobic interactions and multiple weak carbon-hydrogen bonds with all subtypes (Figure A5), but with an additional hydrogen bonding to M3a at position Tyr536 (Figure 4B). For clarity, the 3D model is shown with arecoline forming a definite hydrogen bond to M3a at position Tyr536 (Figure 4C). In short, the overall results suggested that arecoline can bind to pockets of multiple zebrafish mAChRs subtypes and among subtypes, arecoline has the best binding affinities with M3a and M3b. The results from in silico analysis also suggested possible different physiological effects of zebrafish mAChRs subtypes.

### 2.4. Muscarinic Receptor Antagonist Suppressed Locomotion Hyperactivity Induced by Arecoline

Since arecoline displays a broad binding affinity to multiple mAChRs in our docking simulation, it would be difficult to make a solid conclusion for the binding affinity of subtypes of mAChRs. To provide more direct evidence, we conducted another behavioral experiment by treating zebrafish larvae with arecoline and muscarinic acetylcholine receptor antagonists simultaneously. The idea was that locomotion hyperactivity could be diminished when excess antagonists are competing with its receptor (mAChR type 1–4). To this end, muscarinic acetylcholine receptor subtype-specific antagonists such as VU0255035 (M1 antagonist), gallamine triethiodide (M2 antagonist), 4-DAMP (M3 antagonist), and tropicamide (M4 antagonist) were used in the antagonist co-exposure experiment to identify which mAChR subtypes are the major arecoline binding target to alter locomotor activity in zebrafish larvae. Based on the docking score of arecoline, we chose to use these four receptors to study the potential mechanism. According to the molecular docking result, these four receptors (mAChR type 1–4) showed a higher binding score with arecoline, so the importance to study these antagonists was more necessary compare to the other antagonist that has a lower molecular docking score.

The results showed that all mAChRs antagonists tested (M1 to M4) could reduce the hyperactivity phenotype triggered by arecoline. We found that VU0255035, an M1-specific antagonist, could significantly reduce the locomotor activity of larvae locomotor. Interestingly, VU0255035 could decrease the total distance both in light and dark cycles after exposure to arecoline in zebrafish larvae. Even the VU0255035 treatment alone could induce hypoactivity in zebrafish larvae (Figure 5A,E). Furthermore, gallamine (M2-specific antagonist)-treated larvae showed a similar result that the locomotor activity of zebrafish larvae was declined in all of the concentration groups, except in the lowest concentration group (Figure 5B,F). The M3-specific antagonist, 4-DAMP, significantly alleviates the locomotion hyperactivity triggered by arecoline both in dark and light cycles (Figure 5C,G). Results from the M4-specific antagonist, tropicamide, were consistent with other antagonist receptors where tropicamide dampened the locomotor hyperactivity triggered by arecoline (Figure 5D,H).

Interestingly some antagonist-treated larvae, i.e., 4-DAMP treated larvae showed a higher locomotion activity. This could be due to the fact that 4-DAMP is known to act as a non-selective antagonist to muscarinic receptor type 1–4 [40]. Thus, the arecoline-treated larvae still exhibited a higher locomotion activity compared to the antagonist-treated larvae. Taken together, our antagonist co-exposure experiments clearly show the arecoline function as a potent locomotion hyperactivity-stimulating chemical that can activate multiple mAChRs in zebrafish larvae. It is worth noting that throughout exposure to arecoline and mAChR antagonist experiments, there was no mortality or morphological abnormality observed from both groups (Figure A1A–F). In this experiment, some antagonist-treated larvae showed the affected locomotion activity. This means that these antagonists could cause hyperactivity or hypo-activity. These results proved that antagonist compounds could bind with acetylcholine receptors to prevent them from bonding with arecoline.

## 3. Discussion

The robust physiological effects after chewing betel nut have been proved to associate with the alkaloids contained in betel nut. There are four major alkaloids found in betel nut, including arecoline, arecaidine, guvacoline, and guvacine [41]. Among all these alkaloids, arecoline is the most abundant. They constitute from 85–95% of total alkaloids content in the betel nut [5]. Arecoline is associated with euphoria, psychoactive effect, and other effects from betel nut [4,41]. Previous arecoline related studies have mainly investigated toxicological and developmental toxicity. However, it falls short on the potential adverse effects of arecoline in vertebrate models such as zebrafish (in vivo) and behavioral activity. The present study is the first case to evaluate the possible mechanism behind arecoline in affecting the locomotion activity of zebrafish utilizing neuromuscular pathway, and the first case to report a wide range of mAChRs evoked by this drug.

In a study done by Sun et al., it was demonstrated that mice treated with arecoline at a dose of 0.25 to 1 mg/kg for 5 min showed suppression of locomotor activity in a dose-dependent manner [42]. Furthermore, another study demonstrated a decreased locomotor activity when arecoline was given to nzb/b1nj and c57bl/6nnia mice at a dose of 0.64 to 2.5 mg/kg, while an increased locomotor activity was shown when arecoline was given at a high dose of 5.0 to 20.0 mg/kg [43]. Meanwhile, another study in mice showed a different result. Here, they showed that 0.25 to 1 mg/kg of arecoline can lead to a slight enhancement of locomotor activity in mice by promoting spontaneous motor activity [44]. One possibility for the discrepancy of previous studies seen in mice might be contributed by the exposure time wherein the mice experiment, they were treated with arecoline acutely for 3–5 min and also 3 weeks for chronic studies [42,45]. In addition, there is a possibility that the abnormalities in zebrafish larval photomotor response is related to the alteration of the action potential since this response was closely related to the neurons and muscle cells [46]. Therefore, it is intriguing to study this matter in future studies. 

The experiment done by Peng et al. demonstrated that zebrafish treated with arecoline (0.001%, 0.01%, 0.02%, and 0.04%) at an early stage showed a swimming impairment with body length shortening, myosin protein accumulation, and adaxial muscle fibers disorganization [3]. Meanwhile, our current study showed an opposite result. Arecoline-treated larvae demonstrated hyperactivity without any developmental retardation (Figure A1), which indicated that the hyperactivity did not influence any developmental defect such as muscle disorganization. The inconsistency between our experiments with the previous study might be attributed to the differences in the exposure protocols. Peng et al. exposed embryonic zebrafish with arecoline at 4 to 24 hpf of embryogenesis [3], which is a critical stage for motor functions and embryonic maturation [47]. In addition, the segmentation and pharyngula periods are also important stages for zebrafish embryos to develop their neuromeres, primary organogenesis, and also muscle fibers [48]. When arecoline disrupted zebrafish embryonic development, it could lead to an impairment of locomotor activity. Such a notion is supported by a study showing that exposure to arecoline in the early stage can also disrupt zebrafish embryo motor development and their maturation during embryonic development [49]. On the contrary, in our study, zebrafish larvae were exposed to arecoline from 96 hpf onwards to 120 hpf, when organogenesis and morphogenesis were completed. Arecoline exposure in this period did not seem to affect the development of zebrafish larvae as the morphogenesis already had been completed. 

To investigate the time effect of exposure, we tested different exposure durations of arecoline in zebrafish larvae for either 30 min of exposure (short-term) or 24 h of exposure (long-term). Results showed that both short-term and long-term arecoline exposure protocols can trigger locomotion hyperactivity in zebrafish larvae, however severe effects were observed in different cycles between these two protocols (Figure 2). The receptor binding level may play a role in this phenomenon. In a previously published work by Mickey et al., chronic exposure of isoproterenol in rats showed a decreased binding incident between the compound and neuron receptor [50]. There have been other precedents that also showed neuron receptor binding can be affected by arecoline or other compounds that resulted in altered behavior or neuron receptor binding level in the brain or muscle [51,52]. A higher hyperactivity level shown by acutely exposed larvae in the current study could result from a higher binding incident between arecoline with muscarinic and dopaminergic neuron receptors [53]. In addition, a similar phenomenon was also well-known with ethanol exposure, which was used as a positive control in the present study (Figure A4 in Appendix A). Both acute and chronic exposures to ethanol caused anxiety-like behavior in a biphasic manner. Acute exposure to ethanol caused hyperactivity, while chronic exposure reduced locomotion activity [34,54,55]. Another published study also supported the result in the present study, differences in time of exposure to anxiolytic compounds would likely produce different effects [55,56,57]. Taken together, we suggest that exposure time and dosage should be carefully addressed for the arecoline bioactivity test.

The severe hyperactivity effect of arecoline made us question how exactly they affect the behavior of larvae zebrafish at molecular level. Our suspicion leads to mAChRs and arecoline connections. We have done the molecular docking analysis to figure out which muscarinic receptor display the strongest affinity to arecoline ligand. From our docking results, we found that all zebrafish endogenous muscarinic receptor subtypes have high affinities with arecoline, except for the M5 receptor subtype. M1–M4 receptor subtypes have a strong affinity with arecoline (Table 1), especially the M2 and M3 receptor subtypes that play a role in the central neuron system (CNS) [58,59]. It is convincible that the hyperactivity shown by larvae zebrafish in this experiment may be influenced by mAChRs such as M2 and M3 thereby altered zebrafish locomotor activity [60]. Furthermore, our result is also consistent with the previous study showing the arecoline effect as a psychoactive and body excitability promotor. The psychoactive effect of arecoline was indicated by its effect in affecting the behavior of an organism, which is hyperactivity behavior in this case [61]. Meanwhile, its effect as a body excitability promotor was shown by the elevated levels of burst movement and rotation count endpoints which are related to the capability of the organism body into action or a state of excitement [2]. Furthermore, arecoline has been proven to function as a mAChR agonist and could bind with multiple mAChRs [26,62,63,64]. They are capable of stimulating the body into action or a state of excitement (body excitability) and cause hyperactivity by stimulating the mAChRs [41,60]. This alkaloid has a robust effect since it can directly activate mAChRs in the brain since arecoline can pass the blood-brain barrier (BBB) [65]. In addition, arecoline also affects the central nervous system, such as activity enhancement and the condition of responding to certain stimuli by morphine in mice (sensitization) [65,66]. All these experiments support our hypothesis that arecoline could induce hyperactivity by activating mAChRs in zebrafish larvae.

To further prove the hypothesis that arecoline could bind with all the subtype mAChRs, we conducted rescue experiment by incubating zebrafish larvae simultaneously with arecoline and selective mAChR (type 1–4) antagonists. Then, we studied whether hyperactivity caused by arecoline on larvae zebrafish could reverse back to normal level. VU0255035 (type 1 mAChR antagonist), gallamine triethiodide (type 2 mAChR antagonist), 4-DAMP (type 3 mAChR antagonist), and tropicamide (type 4 mAChR antagonist) were used to validate this hypothesis. According to the co-exposure experiment, we found that zebrafish larvae co-exposed to arecoline and mAChR antagonists showed a significantly lower level of locomotor activity compared to the arecoline-only treated larvae (Figure 5). These co-exposure tests combined with the molecular docking analysis indicate that arecoline could induce hyperactivity via multiple mAChRs. All these antagonist receptors are acting as a competitive receptor to arecoline. Previous experiments showed that VU0255035, gallamine triethiodide, 4-DAMP, and tropicamide are competitive antagonists that bind to the same site as an agonist (arecoline), but do not activate it [67,68,69,70]. Thus, they blocked the arecoline biological effects by preventing it from binding with the mAChRs without activating the mAChR function. We conclude that the blockage of arecoline by the mAChR antagonist causing the larvae zebrafish locomotor activity reduced. The present study was in line with previous studies that show that arecoline is actively producing the responses in the central nervous system by acting as a muscarinic agonist with broad activity at M1, M2, M3, and M4 type receptors [16,71]. Muscarinic receptor type 1 (M1) has been reported to be mostly expressed in the forebrain and midbrain that play an important role in motor control [72], which corroborates our result that arecoline treated larvae would exhibit hyperactivity by activating M1. Moreover, selective mAChR M1 antagonists (VU0255035) that were used in the co-exposure experiment, have a potential role in motor control, especially the potential utility in the treatment of move disorders [73,74]. Furthermore, this result also demonstrated that arecoline could bind with type 2mAChR to promote hyperactivity by activating muscarinic receptors and affect the CNS. In a previous study, gallamine triethiodide has been reported to antagonize the effect caused by acetylcholine [75,76]. Studies also suggested that muscarinic receptor antagonists such as gallamine may inhibit agonist-mediated effects [77,78]. In addition, earlier findings showed that gallamine acted as a muscarinic antagonist with a high affinity to M2 receptors [79,80]. The muscarinic receptor has been suggested by several studies as the target against Parkinson’s disease, as mAChR could induce spontaneous movements [81]. Tropicamide itself shows a moderate binding selectivity as a mAChR M4 antagonist [82] and has been used to assess the role of muscarinic receptor subtype function [83,84]. In addition to the fact that antagonist receptors could reduce the hyperactivity caused by arecoline, they also showed some interesting results. The VU0255035 treatment alone seems to cause a lower locomotor activity (Figure 5A,E). This phenomenon is similar to a previous study in mice when mice were treated with VU0255035 exhibited a lower total distance in the locomotor activity test compared to the control [85]. Another interesting finding happens in 4-DAMP treated larvae. 4-DAMP treated zebrafish showed a higher locomotion activity (Figure 5C). This phenomenon may be related to the ability of 4-DAMP to bind with other muscarinic receptors other than M3. Previous studies have found that 4-DAMP has an affinity to M1, M2, M3, and M4 muscarinic receptor subtypes. Thus, they could act as selective receptor antagonist to M2 and act as non-selective receptor antagonist to other subtypes, so the function of the mAChR is not fully occupied by the antagonist and blocked [40,69]. However, a further study is needed to ascertain the hypothesis. Using both in silico molecular docking and in vivo antagonist co-exposure tests, we demonstrated that locomotion hyperactivity triggered by arecoline in zebrafish is mediated by multiple mAChRs, at least M1 to M4. The blockage of arecoline by the antagonist receptor strengthens our hypothesis that arecoline activates multiple mAChRs to cause hyperactivity.

## 4. Materials and Methods

### 4.1. Zebrafish Locomotion Assay to Evaluate Arecoline Bioactivity Workflow Overview

Arecoline was administered acutely (30 min) and chronically (24 h) to observe the effect of arecoline after different time exposures. The acute exposure was conducted 30 min prior to the locomotion activity test, which was on 120 hpf of zebrafish larvae. Meanwhile, for the chronic exposure, zebrafish larvae were subjected to arecoline on 96 hpf, and continued with the locomotion test on 120 hpf [3,86]. After the drug treatment, larvae were transferred to a 48-well plate filled with the drug treatment solution, and their locomotor activity was recorded and tested in a commercial instrument ZebraBox for locomotion tracking and quantification (Figure 6).

### 4.2. Zebrafish Maintenance

Wild type AB strain adult zebrafish (*Danio rerio*) were maintained in a recirculating aquatic system at 28.5 °C with a 10/14 h dark/light cycle, according to the standards. Circulating water in the aquarium was filtered by reverse osmosis (pH 7.0–7.5). The zebrafish were fed twice a day either with lab-grown brine shrimp or dry food. Maintenance and routine culture for the zebrafish were based on the method described by Avdesh et al. [87]. Dry food was obtained from Taiwan Hung Kuo Industrial Co., Ltd., Taiwan. The dry food type is granules and pellets to improve water solubility and to keep the water quality normal. After each crossing, embryos were collected, rinsed, and raised in sterile water with methylene blue (0.00001%) to act as a fungicide, and pH was adjusted to 7.2 with 0.25 ppt of salinity in the temperature-maintained chamber with 28.5 °C [88,89,90,91]. The fish were maintained in a healthy condition and free of any signs of infections and were used according to the guidelines for the care and use of Laboratory Animals by CYCU. All procedures in the present study were approved by the Animal Ethics Committee of the Chung Yuan Christian University (Approval ID 107030).

### 4.3. Arecoline and Muscarinic Acetylcholine Receptor Antagonist Treatment

The water-soluble form of arecoline hydrobromide was purchased from Sigma (Cat# AR-25013, St Louis, MO, USA). For the zebrafish larval locomotion tests, healthy 96 hpf zebrafish larvae were separated into groups of 48 animals. Larvae were exposed to a nominal concentration of 0 ppm (part per million) (control group), as well as concentrations of 0.001, 0.01, 0.1, and 1 ppm either for the short-term or long-term exposure. To study the arecoline antagonist receptor effect, zebrafish larvae aged at 96 hpf were co-treated with mAChR antagonists and arecoline for 24 h until 120 hpf. The locomotor activity was tested.

Muscarinic acetylcholine receptor antagonists were used for the co-exposure experiment. After the molecular docking experiment, based on the docking score we chose the muscarinic antagonist receptor. The mAChR M1 antagonist of VU0255035 was purchased from Bide Pharmatech (Shanghai, China) with a catalog number of BD291918. The mAChR M2 antagonist of gallamine trithiodide was purchased from Aladdin (Shanghai, China) with a catalog number of G129967. The mAChR M3 antagonist of 4-DAMP was purchased from the Toronto research center (Toronto, ON, Canada) with a catalog number of D136400. Lastly, the mAChR M4 antagonist of tropicamide was purchased from Macklin (Shanghai, China) with a catalog number of A800295. Healthy embryos were raised in a 9 cm Petri dish with sterile water until 96 hpf. Later, zebrafish larvae were divided into four groups consisting of 48 larvae each group. Every group was exposed to mAChR subtype-selective antagonists with a concentration of either 0 for the control group or 0.001, 0.01, 0.1 or 1 ppm, respectively. Each concentration of mAChR antagonists was combined with arecoline at the same concentration of mAChR antagonist. Both arecoline and the mAChR antagonist were administered to larvae zebrafish simultaneously. 

### 4.4. Larvae Locomotion Tracking

After incubation with arecoline, zebrafish larvae were transferred and placed into wells of a 48-well transparent plastic plate, one larva in a single well, with 800 µL of 0.001, 0.01, 0.1 or 1 ppm arecoline in each well, and for a co-exposure experiment together with each mAChR antagonist. Then, larvae zebrafish were tested for the locomotion activity mostly within the morning until afternoon (10:00 a.m. to 16:00 p.m.). Plates were placed into the Zebrabox (Viewpoint, Civrieux, France) for locomotion tracking and quantification. The Zebrabox is an isolated recording device with a camera and infrared light-emitting based on where the light was controlled. The ViewPoint system, a video tracking software (Viewpoint, Civrieux, France), was set in tracking mode to record and measure individual zebrafish larval activity. Larvae were first habituated in the dark for 30 min without recording followed by 80 min of recording. The 80-min recording was divided into 4 × 20 min cycles of alternating light (L) and dark (D) periods of 10 min each. Observations were recorded for total distance swam. Swim speed thresholds were set based on the previous study and used to define three different speed thresholds. These speeds including bursting (>2.0 cm/s), which were short, intermittent, and powerful bouts of activity, cruising (0.5 > s < 2.0 cm/s), covering most of the commonly measured larval speeds, and freezing (<0.5 cm/s) during which larvae displayed minimal activity [38]. In addition, the ViewPoint system was also set in quantification and rotation modes, from the recorded videos, to obtain the total burst movement count and total rotations, respectively. For total burst movement counting, the applied thresholds were 20 pixels or more for bursting and less than 5 pixels for freezing. Meanwhile, in the rotation count, clockwise and counterclockwise rotations per minute were counted throughout the test. The thresholds were adjusted based on the minimum diameter (5 mm) and 60° of back angle. Any rotation with a greater value than the minimum diameter and back angle will be counted as one rotation count. All results were binned into one-minute intervals, therefore resulting in 80 data points. Furthermore, to measure larvae swimming responses to a sudden change in light condition, a Larvae photo motor response (LPMR) was observed. LPMR for each photoperiod transition (three light and three dark responses) was calculated as the change in mean distance travelled (in cm) between the last minute of an initial photoperiod and the first minute of the following period.

### 4.5. Structure-Based Molecular Simulation for Arecoline and Muscarinic Acetylcholine Receptor Binding

To investigate the interaction between the inspected arecoline with different subtypes of mAChR, we performed a structure-based molecular simulation study using homology modeling and molecular docking. The in silico work was done on an Asus personal computer with Intel^®^ Core^TM^ i7 2.67 GHz processor, running Windows 7, using Modeller Software v9.20 [92] and Discovery Studio 3.0 (DS 3.0; Discovery Studio Modelling Environment, Accelrys Software Inc, San Diego, CA, USA) [93]. To construct a 3D receptor structure, we first retrieved the target sequence from UniProt (https://www.uniprot.org/ (accessed on 1 January 2021)) and the most relevant homologous structure was used as a template. By sequence alignment between the receptor and template, 3D protein structure models were constructed to evaluate arecoline binding to the virtual receptor. We also searched for potential binding cavities of the receptor through arecoline structure optimization and conducted molecular docking for the ligand-receptor binding simulation. We built the three-dimensional model for each subtype of zebrafish mAChRs by homology modeling using specific amino acid sequences (UniProt codes: M1a: A0A140LG95, M2a: B3DKN8, M3a: X1WHZ7, M4a: E7F3U8, M5a: B3DJA3, M2b: F8W634, M3b: U3JAM0, M5b: A0A2R8RMF6) to locate homologous sequences in the Protein Data Bank. However, we are unable to discover M1b and M4b mAChR homologs since the sequence of *chrm1b* and *chrm4b* could not be found in either Uniprot, NCBI or Z-Fin. Instead, *chrm1a* and *chrm4a*, the other duplicate genes, were used as the template for the molecular docking. The NCBI blastp tool was used to identify the best template structures. We identified eight templates (PDB id: M1a: 6OIJ, M2a: 6OIK, M3a: 4DAJ, M4a: 5DSG, M5a: 6OIJ, M2b: 5ZK8, M3b: 4U15, M5b: 4U14) for eight subtypes of zebrafish endogenous mAChR by homology modeling. For each subtype, the best model out of eight generated structures was carefully selected by the three protein health scoring functions of DOPE [94], and GA341 [95], implemented in the homology modeling software, Modeller [92]. From 10 homology models that were set to be generated by Modeller, eight homology protein structures were made. GA341 uses the homology sequence identity between the template and the model as a judge. GA341 ranges from 0.0 (worst) to 1.0 (native-like). DOPE is “Discrete Optimized Protein Energy”, a statistical potential optimized for the health of protein. The lower DOPE is, the more stable the model is. Next, to analyze if arecoline can bind to zebrafish mAChRs, we conducted molecular docking using the docking module in Discovery Studio 3.0, LigandFit [96]. A cavity searching method, eraser algorithm [30], located the best binding pocket for docking in each built 3D model of the mAChR subtypes, as shown in Figure 6B. Arecoline was optimized with mmff94 force field (Chem3D) until the energy is converging to the root-mean-square-deviation (RMSD) gradient within 0.05 kcal mol^−1^ Å^−1^, the structure reaches the lowest point. Partial charges of all atoms within the receptor were assigned, and all hydrogens were restored based on the CHARMm force field [31]. The dock scoring function measured the interaction energy between ligand pose and receptor. Any docking pose should fit into the generated cavity, and the higher dock score indicated the stronger binding affinity. In summary, the best docking poses of arecoline buried in mAChR subtypes were visualized in the graphics environment of UCSF Chimera 1.13.1 [32].

### 4.6. Statistical Analysis

The experimental values were compared between control and treated groups, except when it is otherwise noted. For behavioral results, most of the data were expressed as the median with the interquartile range or 95% CI since the data did not follow a normal distribution. However, several behavioral data with 0 value of median were expressed as the means ± SEM or means with SD to display more representative data. All tests were conducted through either the Mann-Whitney, the two-way ANOVA with Geisser Greenhouse correction, Brown-Forsythe ANOVA continued with Dunnett’s T3 multiple comparison tests, or the Kruskal-Wallis test continued with Dunn’s multiple comparison tests as a follow-up test. For total distance, burst movement, and average rotation, repeated measures two-way ANOVA was used followed by Geisser-Greenhouse correction and Dunnett’s multiple comparison tests. For short-term and long-term comparisons, we used the Mann-Whitney test. Furthermore, for the LPMR test, the data were analyzed using repeated measures two-way ANOVA with Geisser-Greenhouse correction. Meanwhile, for the antagonist experiment, all the data were analyzed using the Kruskal-Wallis test with Geisser-Greenhouse correction followed by Dunn’s multiple comparison test. For the morphology test, significance was tested by the Brown-Forsythe ANOVA test continued with Dunnett’s T3 multiple comparison test as a follow-up test. The non-parametric analysis was applied in the current study since generally, fish behavior data do not meet the assumption of normal distribution [97]. Statistical tests were performed using GraphPad Prism (https://www.graphpad.com/ (accessed on 1 January 2021)).

## 5. Conclusions

Arecoline is a potent compound that affects larvae zebrafish behavior by increasing locomotor activity and alters motor function. The potential role of mAChRs was meticulously illustrated by molecular docking simulation and the antagonist co-exposure experiment, supporting the idea that the locomotion hyperactivity triggered by arecoline is mediated by multiple mAChRs. In a nutshell, our structure-based simulations indicate that arecoline has broad binding efficiency to multiple mAChRs, suggesting a potential influence on binding efficiency or physiological effects. Moreover, the results of the mAChRs-selective antagonist in this study strengthen the hypothesis that arecoline generates hyperactivity via multiple mAChRs in vivo. Data from molecular docking and receptor antagonists complement each other to argue that arecoline can bind with multiple mAChRs (M1–M4) to produce locomotion hyperactivity and any move disorders. We have identified the detailed interactions between arecoline and several mAChRs by the 2D/3D docking approach and identified an additional hydrogen bond formed between zebrafish endogenous mAChR M3a and arecoline. The mAChR M3b also scores a high binding affinity among all the subtypes tested in this study. Taken together, this zebrafish-based assay platform provides clues to explain the potential mechanism and effectivity of the arecoline component in the areca nut on promoting locomotion hyperactivity for the first time.

It is worth noting that the broad binding spectrum nature of arecoline to multiple mAChRs makes it difficult to get crystal clear results based on pharmacology and molecular docking-based approaches. Studies using either new synthesized selective mAchRs antagonists or knocking out specific mAChR subtypes with genome editing tools will provide more solid and promising evidence to the detailed mechanism on mediating locomotor hyperactivity triggered by arecoline in zebrafish. Further studies are called for to explore the expressional territories of mAChR subtypes among diverse tissues and delineate the potential expression-function relationship mediating arecoline stimulus.

## Figures and Tables

**Figure 1 toxins-13-00259-f001:**
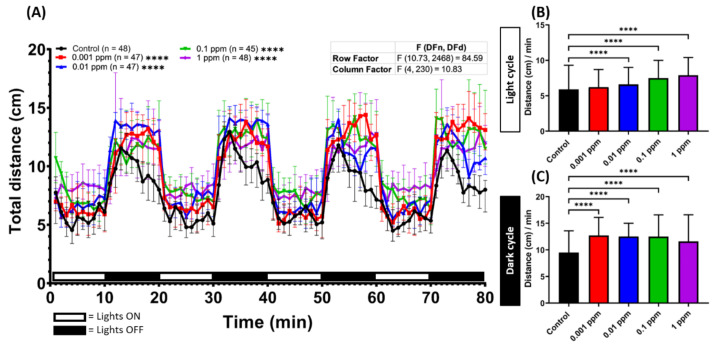
Arecoline induced locomotion hyperactivity in zebrafish larvae. (**A**) Average distance travelled per minute by 120 hpf zebrafish larvae after 1-day exposure of 0 ppm (black), 0.001 ppm (red), 0.01 ppm (blue), 0.1 ppm (green), and 1 ppm (purple) arecoline. The data are expressed as the median ±95% CI and the significance was tested by two-way ANOVA with the Geisser-Greenhouse correction. To observe the main column (arecoline) effect, Dunnett’s multiple comparison test for comparing all treatments with the control was carried out. Total average distance travelled per minute during (**B**) the light and (**C**) dark cycles at different arecoline concentrations was compared. The data are expressed as the median with interquartile range and significance was tested by the Kruskal-Wallis test continued with Dunn’s multiple comparisons as a follow-up test. Each treated group was compared with the control group (*n* = 48 for the control, and 1 ppm arecoline groups; *n* = 47 for 0.001 and 0.01 ppm arecoline groups; *n* = 45 for 0.1 ppm arecoline group, **** *p* < 0.0001).

**Figure 2 toxins-13-00259-f002:**
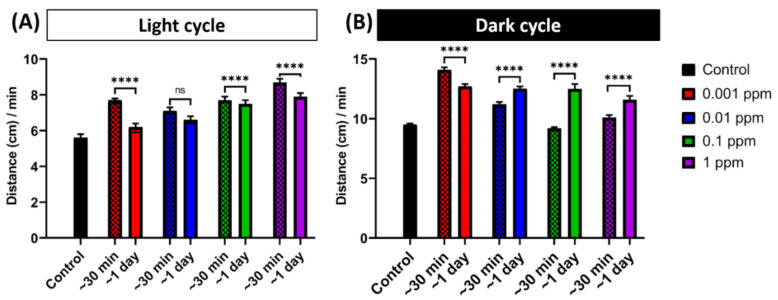
Comparison of short-term and long-term arecoline exposure on promoting locomotion hyperactivity. Arecoline induced a different pattern of hyperactivity behavior after short-term (30-min) and long-term (1-day) exposure in zebrafish larvae. Total average distance travelled per minute by 120 hpf zebrafish larvae during (**A**) light and (**B**) dark cycles after short-term (30-min) and long-term (1-day) arecoline exposure at different dosages were compared. The data are expressed as the median with 95% CI and significances were tested by the Mann-Whitney test. The statistical analyses were conducted between each treated group in every concentration (*n* = 48 for the control, 1-day exposure of 0.01 and 1 ppm, and 30-min exposure of 1 ppm arecoline groups; *n* = 47 for 1-day exposure of 0.001 ppm and 30-min exposure of 0.001 and 0.01 arecoline groups; *n* = 45 for 1-day exposure of 0.1 ppm and 30-min exposure of 0.1 ppm arecoline groups, **** *p* < 0.0001).

**Figure 3 toxins-13-00259-f003:**
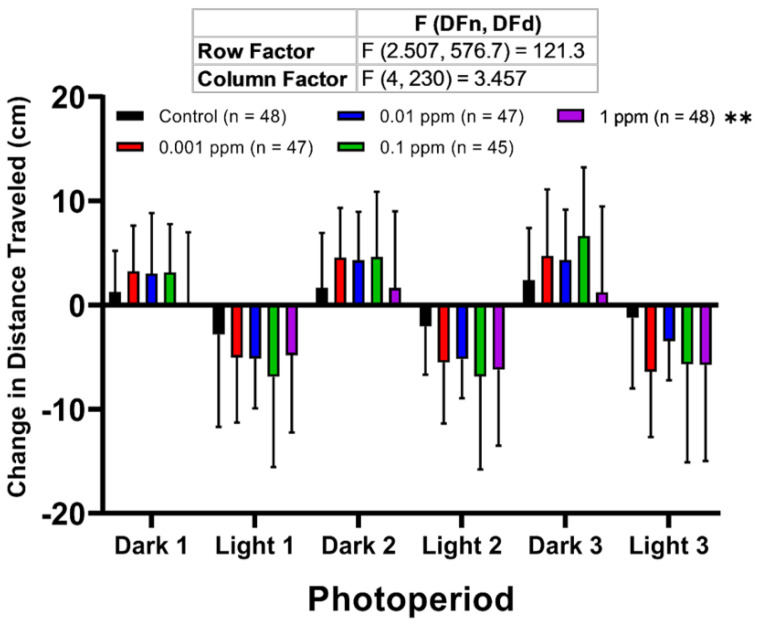
Comparison of the larvae photomotor response (LPMR) for zebrafish larvae after treated with different doses of arecoline. LPMR of 120 hpf zebrafish after 1-day exposure of 0 ppm (black), 0.001 ppm (red), 0.01 ppm (blue), 0.1 ppm (green), and 1 ppm (purple) arecoline. Three LPMRs were measured. The data are expressed as means ± SD of total distance travelled of each following period. Data were analyzed by two-way ANOVA with the Geisser-Greenhouse correction. Each treated group was compared with the control group (*n* = 48 for the control, 0.01 and 1 ppm arecoline groups; *n* = 47 for 0.001 ppm arecoline group; *n* = 45 for 0.1 ppm arecoline group, ** *p* < 0.01).

**Figure 4 toxins-13-00259-f004:**
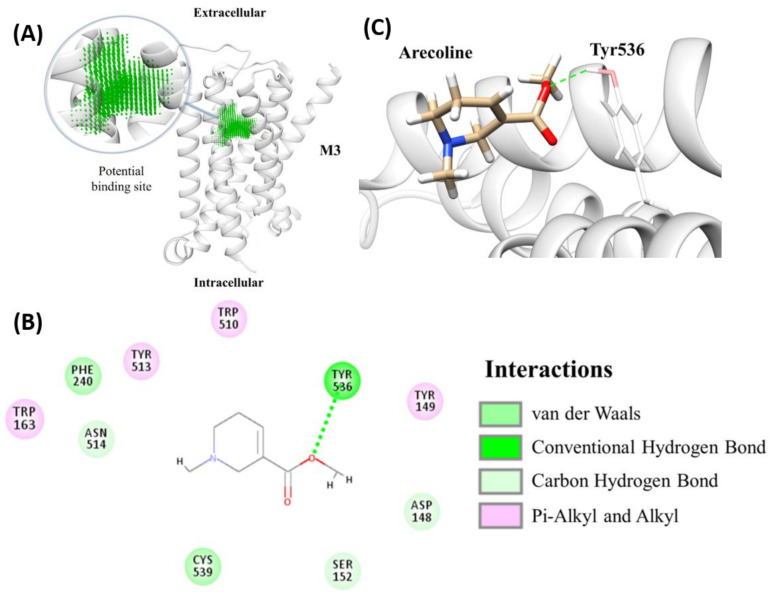
Molecular docking of arecoline with endogenous zebrafish muscarinic acetylcholine receptor (mAChR). (**A**) The identified binding pocket in the middle of the hollow cylinder of the eight sub-types homology modeling structures of mAChR (M3a as an example). Two-dimensional (2D) (**B**) and 3D (**C**) illustrations of interactions between arecoline and endogenous zebrafish mAChR M3a showing the hydrogen bond formation between arecoline and mAChR M3a at position Tyr536 (highlighted by the green dotted line).

**Figure 5 toxins-13-00259-f005:**
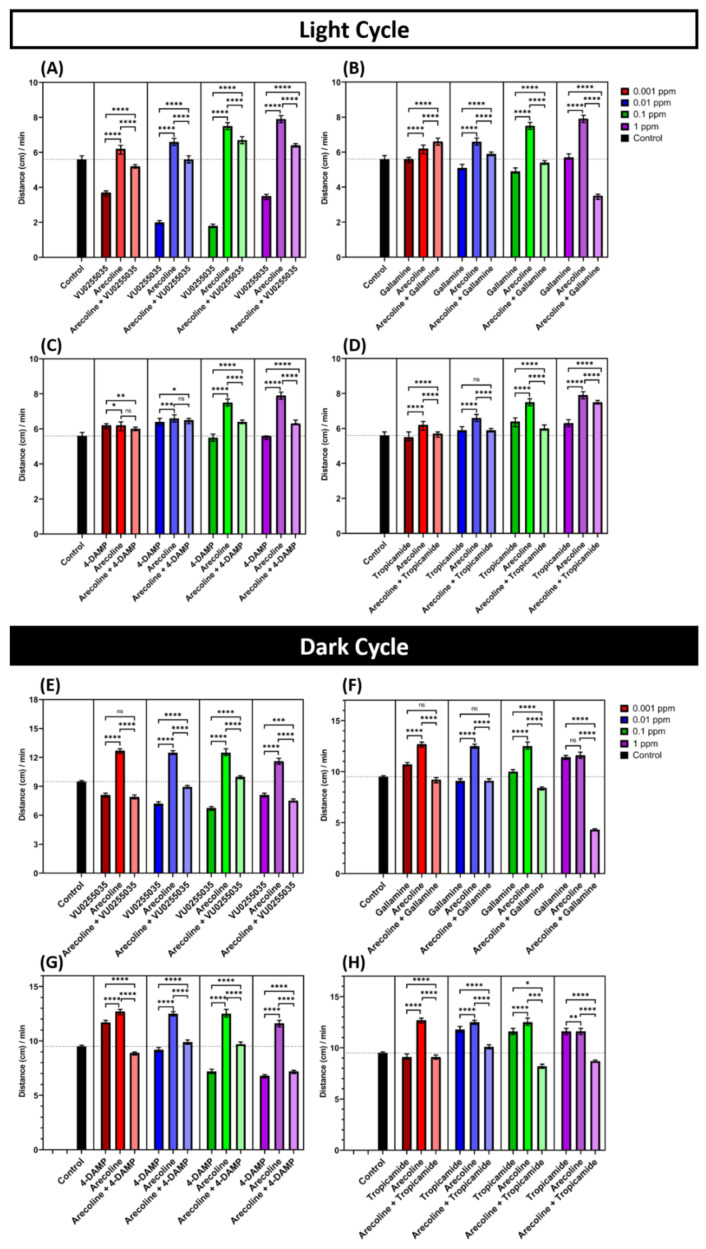
Muscarinic acetylcholine receptor (mAChR) antagonists can reduce the locomotion hyperactivity triggered by arecoline in the light and dark cycles in zebrafish larvae. (**A**–**H**) Total average distance travelled by 120 hpf zebrafish larvae per minute after 1-day exposure of 0.001, 0.01, 0.1, and 1 ppm VU0255035 (mAChR M1-selective antagonist), gallamine (mAChR M2-selective antagonist), 4-DAMP (mAChR M3-selective antagonist), and tropicamide (mAChR M4-selective antagonist), respectively, and their combination with 0.001, 0.01, 0.1, and 1 ppm arecoline in light and dark cycles, respectively. The data are expressed as the median with 95% CI and significance was tested by the Kruskal-Wallis test with Geisser-Greenhouse correction followed with Dunn’s multiple comparison test. The statistical analyses were conducted between each treated group in every concentration (*n* = 48 for all of the groups, except for 0.001 and 0.01 ppm arecoline (*n* = 47), 0.1 ppm arecoline, 1 ppm 4-DAMP, and 0.1 ppm tropicamide (*n* = 45), 0.1 ppm 4-DAMP and 1 ppm of arecoline and tropicamide (*n* = 46), * *p* < 0.05, ** *p* < 0.01, *** *p* < 0.001, **** *p* < 0.0001).

**Figure 6 toxins-13-00259-f006:**
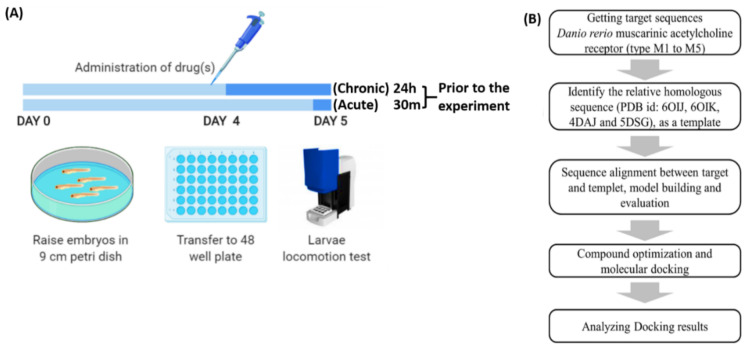
Zebrafish locomotor assay to evaluate the arecoline bioactivity workflow overview. (**A**) Upper panel: Schematic showing the timing for drug administration. Lower panel: Schematic showing the instruments used for drug incubation and locomotion tracking. (**B**) In silico molecular docking pipeline used to explore the binding affinity between arecoline and *Danio rerio* endogenous muscarinic acetylcholine receptors (mAChRs).

**Table 1 toxins-13-00259-t001:** Homology modeling structures of zebrafish muscarinic acetylcholine receptor (chrm) subtypes in this research. # Two subtypes of chrm3a and chrm3b binding to arecoline not only gain the highest dock scores but also have one additional hydrogen bond formation which was not found in other subtype-arecoline interactions.

Gene	Uniprot (ID)	Length (a.a.)	PDB ID/Organism/Identity with *D. rerio*	Dock Score
*chrm1a*	A0A140LG95	465	6OIJ/human (*Homo sapiens*)/74.05%	35.909
*chrm2a*	B3DKN8	495	6OIK/human (*Homo sapiens*)/90.29%	36.896
*chrm2b*	F8W634	466	5ZK8/human (*Homo sapiens*)/82.12%	34.155
*chrm3a*	X1WHZ7	595	4DAJ/brown rat (*Rattus norvegicus*)/81.48%	38.108#
*chrm3b*	U3JAM0	494	4U15/brown rat (*Rattus norvegicus*)/80.59%	38.419#
*chrm4a*	E7F3U8	513	5DSG/human (*Homo sapiens*)/87.50%	34.285
*chrm5a*	B3DJA3	490	6OIJ/human (*Homo sapiens*)/74.91%	26.621
*chrm5b*	A0A2R8RMF6	505	4U14/brown rat (*Rattus norvegicus*)/74.33%	1.839

## Data Availability

The original data and video are available upon request from the authors.

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
