# Peer review of "Pharmaceutical Assessment Suggests Locomotion Hyperactivity in Zebrafish Triggered by Arecoline Might Be Associated with Multiple Muscarinic Acetylcholine Receptors Activation"

_toxins, 2021, doi:10.3390/toxins13040259_

Round 1
Reviewer 1 Report
In this manuscript, the authors have examined the effects of arecoline on zebrafish locomotor activity following short (30min) and long (24hr) exposure in larvae. The authors then use a computational binding assay approach to determine which mAChR subtype(s) arecoline may bind to and has the best affinity. The authors then performed co-exposure/rescue experiments to assess if blocking specific mAChRs will remove arecoline’s effects. The authors identified increased locomotor activity due to arecoline exposure and that arecoline likely exerts this effect by binding to different mAChR types. The study is relevant, however, the document needs a significant amount of work related to clarity of methods and reductions in redundancy prior to publication.
Specific comments:
- line 74-75 states “There has been a very limited number of studies utilize zebrafish larvae, especially in their behaviors.” This statement is not true. Zebrafish larvae are used in a variety of behavioral assessments, particularly in toxicology-type studies similar to this one. The authors should reword this sentence and include relevant publications.
- lines 106-107; the authors state that previous studies show an effect of arecoline on locomotor activity of zebrafish larvae. Two points here: (1) this contradicts what they say previously in the second paragraph of the introduction and (2) this questions the novelty of the current work.
- Why did the authors chose the concentrations of arecoline that they used? Why did the authors choose to expose for 30 minutes or 24 hours only?
- line 124 states ‘Exposure to arecoline was performed on 120hpf’ however on the next page (lines 155-156) and in Figure 1, exposure actually occurred before 120hpf. Please edit the text for consistency.
- Please make your headings in the methods and results consistent. Section 2.3 is ‘Arecoline and muscarinic acetylcholine receptor antagonist treatment’. This correlates with Section 3.4, which has a different heading.
- The term ‘rescue’ is used to refer to the co-exposure experiment. ‘Rescue’ has a very specific meaning in molecular genetic studies. Co-exposure may be the better word here.
- During the co-exposure experiments, what was the specific procedure used? Was arecoline administered first, then the antagonist? Or was the antagonist administered first, then arecoline? Or were they administered simultaneously? Use of the term ‘rescue’ suggests that arecoline was administered first; however, lines 433-434 suggest that the antagonist may have been administered first. Also, the methods (lines 169-170) states that arecoline and each antagonist was administered in the same concentrations. So, if 1ppm arecoline was used was 1ppm of an antagonist also used? Or were the concentrations different? This is important: what is know about how the antagonists work? Are the competing with arecoline for the site on the receptor? Or do they bind at a different site? Competitive vs. noncompetitive actions of the antagonist and arecoline are important should be included in the text and results should be discussed with this in mind.
- Is the LPMR the same as the visual motor behavioral assay (VMR)?
- One striking feature in the graphs is the increased activity during light OFF. This is characteristic of zebrafish larvae. The authors should speak to this, as there has to be activity above this inherent increase to classify the larvae as ‘hyperactive’. This difference is present in all their locomotor measurements.
- The authors analyzed their data using ANOVAs. Were these repeated measures ANOVAs? Repeated measures seem most appropriate since the same larvae were assessed multiple times.
- The first part of section 3.3 is mostly methods and could be condensed here and moved to the methods section.
- line 356 states that ‘we constructed eight homology protein structures’. This contradicts the methods (line 218) which states that 10 were created.
- What is considered a good ‘dock score’? is there a maximum score that can be achieved? It would be helpful to have a frame of reference for these values.
- Table 1 has two dock scores with asterisks. These asterisks seem to highlight these values and not be associated with statistics. It they are just to highlight, I suggest using a different symbol or bold those values to highlight, rather than use asterisks.
- section 3.4 – in the co-exposure experiments, the authors use antagonists for 4 different mAChR subtypes, but only two are highlighted in the table. Why were the other two receptor types selected?
- discussion – this section is long and very redundant with many grammatical errors. The authors should edit this section, making the text more concise (reduce by 10-20%). In addition,
- line 473 gives 0.25-20mg/kg as the concentrations of arecoline used in mice and notes this is “too broad and could lead to different results” (line 474). This is a 100-fold difference. The authors used concentrations of 0.001-1ppm, which is a 1000-fold difference. The concentrations used here are even broader than the indicated study.
- the authors also comment that arecoline may have a U-shaped dose response, but can’t make that determination because ‘only four concentrations were tested’ (line 484-485). However, examination of the figures suggests that the authors may be able to determine is the dose response is linear or not. While they see the trend of increasing effect with increasing arecoline concentration, the responses in Figure 4 during the dark period suggest that intermediate concentrations cause the greatest effect, not the highest concentration. Also, Figure A2 shows hyperactivity (above controls) with low arecoline concentrations while hypoactivity occurred at higher concentrations in dark conditions.
- line 475-476 the authors state that they chose concentrations of arecoline so they would not cause ‘developmental retardation’. Please provide a reference for >1ppm causing developmental retardation. This information should also be moved earlier in the document, as it gives relevance to why the concentrations used were chosen.
- line 524 mentions that arecoline can bind to dopaminergic receptors; however, this is not discussed further nor is there a reference given. How might this information affect the interpretation of your results?
- paragraph between lines 510-536 discusses how different exposure times can cause different results. Was this observed in this study? Hyperactivity was seen after a 30min exposure and after a 24hr exposure, though it did vary depending on if activity was measured during light on or off.
- line 537-538 states “our result is consistent with the previous study showing the arecoline effect as a psychoactive and body excitability promotor”. What tests were performed to determine psychoactive effects? What is the difference between body excitability and hyperactivity?
- What is ‘average rotation’ measuring? Is this turn angle or turning? How might this parameter be affected by the small size of the well in the 48 well plate?
- Within the figures, it is hard to differentiate among the different symbols. The different colors help a lot, though.
Author Response
In this manuscript, the authors have examined the effects of arecoline on zebrafish locomotor activity following short (30min) and long (24hr) exposure in larvae. The authors then use a computational binding assay approach to determine which mAChR subtype(s) arecoline may bind to and has the best affinity. The authors then performed co-exposure/rescue experiments to assess if blocking specific mAChRs will remove arecoline’s effects. The authors identified increased locomotor activity due to arecoline exposure and that arecoline likely exerts this effect by binding to different mAChR types. The study is relevant, however, the document needs a significant amount of work related to clarity of methods and reductions in redundancy prior to publication.
Specific comments:
- line 74-75 states “There has been a very limited number of studies utilize zebrafish larvae, especially in their behaviors.” This statement is not true. Zebrafish larvae are used in a variety of behavioral assessments, particularly in toxicology-type studies similar to this one. The authors should reword this sentence and include relevant publications.
The authors strongly agree with the reviewer’s point. It is true that zebrafish larvae are used in a variety of behavioral assessments, including in toxicology studies which are similar to the present study. Therefore, the mentioned statement had revised to avoid misleading information. In addition, to support the current statement, several relevant prior publications mentioned below were also added in this section, especially in lines 75-77 as the reviewer suggested.
Richendrfer, H., et al., On the edge: pharmacological evidence for anxiety-related behavior in zebrafish larvae. Behavioural brain research, 2012. 228(1): p. 99-106.
Ulhaq, M., et al., Locomotor behavior in zebrafish (Danio rerio) larvae exposed to perfluoroalkyl acids. Aquatic toxicology, 2013. 144: p. 332-340.
Zhang, W., et al., Toxicity assessment of zebrafish following exposure to CdTe QDs. Journal of hazardous materials, 2012. 213: p. 413-420.
2. lines 106-107; the authors state that previous studies show an effect of arecoline on locomotor activity of zebrafish larvae. Two points here: (1) this contradicts what they say previously in the second paragraph of the introduction and (2) this questions the novelty of the current work.
The authors fully understand the reviewer’s point. In responding to the reviewer’s first point, the authors admit that there was a mistake that causes a contradictive statement in the following paragraph. Therefore, the mentioned part in the second paragraph was revised so there is no contradictive statement in the current version manuscript. Furthermore, regarding the novelty of the current work, the authors had added the novelty in the current work, which is the combination of in silico and in vivo approaches. Later, with this combination, the authors proved several evidence suggesting that arecoline could activate multiple receptors and affect zebrafish larval locomotor activity that has not been explored in any previous experiment. In addition, even though a prior study already tested the effect of arecoline in locomotion of zebrafish, it does not reduce the novelty of the present study since, in their study, arecoline was administered to zebrafish embryos from 4 to 24 hours post-fertilization, which is different to the current study’s protocol.
3. Why did the authors chose the concentrations of arecoline that they used? Why did the authors choose to expose for 30 minutes or 24 hours only?
Thank you for the question. The authors chose those particular concentrations based on several prior studies of arecoline’s effect in zebrafish larvae. Some of these studies found that 0.01-0.04% of arecoline caused general growth retardation, morphological deformities, and lower heart rate. Meanwhile, another study found that arecoline in a higher concentration (>1 ppm) has a potentially adverse effect on the animal. Taken together, the current concentrations (0.001-1 ppm) were chosen to make sure that developmental retardation did not interfere with their abnormal locomotion. In addition, these concentrations were much less than the daily dose of arecoline consumed by betel chewers or the dose used for the treatment in Alzheimer’s patients (9.6-61 mg/day). Next, the 30 minutes and 24 hours-exposure times were chosen to represent acute and chronic exposures effect of arecoline based on some previous studies. By using these exposure times, the authors wanted to find out whether a different time of exposure could induce a different effect of arecoline or not since, in some chemicals, such as ethanol, different effects was caused after different time exposure. All of this additional information were added to the manuscript, especially in the Introduction part.
Airhart, M. J., Lee, D. H., Wilson, T. D., Miller, B. E., Miller, M. N., & Skalko, R. G. (2007). Movement disorders and neurochemical changes in zebrafish larvae after bath exposure to fluoxetine (PROZAC). Neurotoxicology and teratology, 29(6), 652-664.
Asthana, S., et al., Neuroendocrine responses to intravenous infusion of arecoline in patients with Alzheimer's disease. Psychoneuroendocrinology, 1995. 20(6): p. 623-636.
Chang, B.E., et al., Developmental toxicity of arecoline, the major alkaloid in betel nuts, in zebrafish embryos. Birth Defects Research Part A: Clinical and Molecular Teratology, 2004. 70(1): p. 28-36.
Dasgupta, R., et al., Ultrastructural and hormonal modulations of the thyroid gland following arecoline treatment in albino mice. Molecular and cellular endocrinology, 2010. 319(1-2): p. 1-7.
Peng, W.-H., et al., Short-term exposure of zebrafish embryos to arecoline leads to retarded growth, motor impairment, and somite muscle fiber changes. Zebrafish, 2015. 12(1): p. 58-70.
4. line 124 states ‘Exposure to arecoline was performed on 120hpf’ however on the next page (lines 155-156) and in Figure 1, exposure actually occurred before 120hpf. Please edit the text for consistency.
The authors appreciated the detailed correction. It is true that there were some mistyping in the mentioned statements that made some inconsistencies regarding the time of arecoline exposure. Therefore, to avoid confusion, the sentence in line 124 was corrected according to the correct method, which is the arecoline exposure occurred before 120 hpf.
5. Please make your headings in the methods and results consistent. Section 2.3 is ‘Arecoline and muscarinic acetylcholine receptor antagonist treatment’. This correlates with Section 3.4, which has a different heading.
Thank you for the suggestion. The authors understand that sections 2.3 and 3.4 contain similar content to each other. However, from the author’s perspective, these sections have their own specific information that if they have the same heading, it could eliminate some important information from each section. In the material and methods section, section 2.3 is intended to give the information regarding the arecoline and its receptor antagonist exposures to zebrafish larvae while in the results part, section 3.4 was intended to emphasize that its receptor antagonist was able to reduce the hyperactivity caused by arecoline treatment. Taken together, the authors believe that it is more appropriate to give a different heading for each section.
6. The term ‘rescue’ is used to refer to the co-exposure experiment. ‘Rescue’ has a very specific meaning in molecular genetic studies. Co-exposure may be the better word here.
The authors strongly agreed with the reviewer. The word ‘rescue’, indeed, has a very specific meaning in molecular genetic studies. Furthermore, even though the present study wants to emphasize that the muscarinic acetylcholine receptor antagonists could suppress the zebrafish locomotion hyperactivity caused by arecoline treatment, the usage of this term in the manuscript might not suitable as the reviewer pointed out. Therefore, since in the present study the incubations of arecoline and all of the muscarinic acetylcholine receptor antagonists were conducted simultaneously, the word ‘co-exposure’ may be more appropriate than ‘rescue’. Based on this consideration, the word ‘rescue’ in the whole manuscript was revised to ‘co-exposure’, according to the reviewer’s suggestion.
7. During the co-exposure experiments, what was the specific procedure used? Was arecoline administered first, then the antagonist? Or was the antagonist administered first, then arecoline? Or were they administered simultaneously? Use of the term ‘rescue’ suggests that arecoline was administered first; however, lines 433-434 suggest that the antagonist may have been administered first. Also, the methods (lines 169-170) states that arecoline and each antagonist was administered in the same concentrations. So, if 1ppm arecoline was used was 1ppm of an antagonist also used? Or were the concentrations different? This is important: what is know about how the antagonists work? Are the competing with arecoline for the site on the receptor? Or do they bind at a different site? Competitive vs. noncompetitive actions of the antagonist and arecoline are important should be included in the text and results should be discussed with this in mind.
Thank you for pointing out this matter. First of all, regarding the administration of arecoline and acetylcholine receptor antagonists, the authors fully aware that the usage of the word ‘rescue’ caused confusion in understanding the manuscript. Therefore, as is already mentioned above, this word had changed to ‘co-exposure’ throughout the whole manuscript since it is more suitable to the current experiment. Based on this change, it means that the arecoline was co-exposed with acetylcholine receptor antagonists, which implies that they were both administered to zebrafish larvae simultaneously. In addition, the statement in lines 433-434 that suggested that the antagonist may have been administered first had also been revised accordingly. Secondly, regarding the concentration ratio between arecoline and acetylcholine receptor antagonists in lines 169-170, the authors strongly agree about the importance of this matter to address some intriguing results, as the reviewer mentioned above. In the present study, the same concentration of arecoline and acetylcholine receptor antagonists was applied in this study, thus, as the reviewer asked, the 1 ppm concentration means that both arecoline and antagonist receptor concentrations were 1 ppm. In addition, in terms of the antagonist’s mechanism possibility, the antagonist muscarinic receptors used in this experiment act as a competitive antagonist. Therefore, all of the compounds will bind with the same site as arecoline and prevent arecoline to bind with muscarinic acetylcholine receptor. However, arecoline was still found to affect zebrafish larval locomotor activity, thus, it indicates that arecoline could bind with multiple receptors. All of this essential information based on the reviewer’s suggestion was added to the manuscript, specifically in the discussion section.
8. Is the LPMR the same as the visual motor behavioral assay (VMR)?
Thank you for raising this question. Actually, based on the definition of each term described by several prior studies mentioned below, LPMR and VMR are not the same. While LPMR was used to measure the larval swimming responses toward a sudden change in light condition, visual-motor response (VMR) behavioral assay was used to analyze how genetic or environmental manipulations alter neurological function. Moreover, based on the previous experiment, a VMR assay was performed to analyze neural circuits in the zebrafish central nervous system (CNS) and elucidate the pathogenic mechanism in zebrafish disease models by trying different ranges of wavelength. On the other hand, the current experiment aimed to observe the effect of arecoline and acetylcholine receptor antagonists on larval swimming responses toward a sudden change in light (light on (light) and light off (dark)) condition. Therefore, the authors believed that LPMR was more appropriate to use in the present study.
Burton, C. E., Zhou, Y., Bai, Q., & Burton, E. A. (2017). Spectral properties of the zebrafish visual motor response. Neuroscience letters, 646, 62-67.
van Woudenberg, Anna Beker, et al. "A category approach to predicting the developmental (neuro) toxicity of organotin compounds: the value of the zebrafish (Danio rerio) embryotoxicity test (ZET)." Reproductive toxicology 41 (2013): 35-44.
9. One striking feature in the graphs is the increased activity during light OFF. This is characteristic of zebrafish larvae. The authors should speak to this, as there has to be activity above this inherent increase to classify the larvae as ‘hyperactive’. This difference is present in all their locomotor measurements.
The authors appreciate the reviewer for pointing out this important matter. As the reviewer stated, it is true that the increased activity of zebrafish larvae during the dark cycle (light OFF) is one of their important characteristics, and a higher level of locomotion is observed in all of the arecoline-treated group’s locomotion are classified as ‘hyperactive’ behavior. Therefore, this important information regarding the natural characteristic of zebrafish larvae and the abnormal activity of the arecoline-treated zebrafish larvae was added and discussed in the manuscript, especially in the results part, in section 2.1.
10. The authors analyzed their data using ANOVAs. Were these repeated measures ANOVAs? Repeated measures seem most appropriate since the same larvae were assessed multiple times.
Thank you for the question. Here, the authors used repeated-measures ANOVA since the zebrafish larval locomotion activities were measured multiple times (every 1 minute for 80 minutes). As the reviewer mentioned, the repeated measures ANOVA is the most appropriate test because in the dataset, each row represents a different time point, thus, matched values are stacked into a subcolumn. In addition, to enhance the clarity of the statistical analyses part, this additional information was added to the manuscript, specifically in the materials and methods part, in section 2.6.
11. The first part of section 3.3 is mostly methods and could be condensed here and moved to the methods section.
The authors appreciate the suggestion. After conducting a further evaluation regarding this matter, the authors agree that the first paragraph of section 3.3, which mostly discussed the process of molecular docking of arecoline with eight subtypes of endogenous zebrafish muscarinic acetylcholine receptors (mAChR), was more suitable to be moved to the methods section. Therefore, as the reviewer suggested, this paragraph had moved to the materials and methods section, specifically in section 5.3. In addition, in this updated version, this matter had also condensed to avoid a lengthy explanation according to the reviewer’s suggestion.
12. line 356 states that ‘we constructed eight homology protein structures’. This contradicts the methods (line 218) which states that 10 were created.
Thank you for the correction. Actually, there was a typing mistake regarding the making of homology protein structures. In fact, the correct statement is from the 10 best-generated homology models that were set to be generated by Modeller software, only eight homology protein structures were generated since the other two sequences, which were chrm1b and chrm4b, could not be found in either Uniprot, NCBI, or Z-Fin. This correct information was added to the manuscript, specifically in section 5.5 to avoid confusion.
13. What is considered a good ‘dock score’? is there a maximum score that can be achieved? It would be helpful to have a frame of reference for these values.
The authors appreciated the question. Actually, there is no minimum or maximum score in considering whether a docking score is a good ‘dock score’ or not. Here, the docking score was used as evidence to show the capability of arecoline to bind and has an affinity with a muscarinic receptor and even though it should not be used to select strong binders, it helped the authors to take the decision in eliminating the ‘bad’ binders. Based on this consideration, the four types of muscarinic acetylcholine receptor, which were type 1-4, were selected since they had a higher dock score than acetylcholine receptor type 5. Therefore, with this evidence, the co-exposure experiment was carried out afterward in order to study the possible mechanism of arecoline in affecting the locomotor activity of zebrafish larvae. This important information regarding the docking score was added to the manuscript, specifically in section 2.3.
14. Table 1 has two dock scores with asterisks. These asterisks seem to highlight these values and not be associated with statistics. It they are just to highlight, I suggest using a different symbol or bold those values to highlight, rather than use asterisks.
Thank you for the suggestion. As the reviewer mentioned, it is true that the asterisks in Table 1 were used to highlight the dock score values of some genes and they are not associated with statistics, thus, the usage of these asterisk in this table might cause confusion to the readers. Therefore, to avoid this problem, the symbol to highlight the dock scores in Table 1 was changed to ‘#’. In addition, the table’s caption was also revised accordingly.
15. section 3.4 – in the co-exposure experiments, the authors use antagonists for 4 different mAChR subtypes, but only two are highlighted in the table. Why were the other two receptor types selected?
The authors appreciated the reviewer for asking this important question. Actually, the highlights displayed in Table 1 are not intended to show the highest dock scores of two subtypes of chrm3a and chrm3b binding to arecoline, but rather to point out that they possess one additional hydrogen bond formation that was not found in other subtype-arecoline interactions. Therefore, these highlights are not really related to the authors’ decision in selecting also the other two receptor types. As mentioned in the previous point, the selection of these four receptor types was based on their high dock score levels, which is not observed in subtype 5.
16. discussion – this section is long and very redundant with many grammatical errors. The authors should edit this section, making the text more concise (reduce by 10-20%). In addition,
a. line 473 gives 0.25-20mg/kg as the concentrations of arecoline used in mice and notes this is “too broad and could lead to different results” (line 474). This is a 100-fold difference. The authors used concentrations of 0.001-1ppm, which is a 1000-fold difference. The concentrations used here are even broader than the indicated study.
Thank you for the correction. It is true that the present study used broader concentrations than the mentioned previous experiment. Therefore, to avoid some contradictive statements, the paragraph that refers to the statement was revised.
b. The authors also comment that arecoline may have a U-shaped dose response, but can’t make that determination because ‘only four concentrations were tested’ (line 484-485). However, examination of the figures suggests that the authors may be able to determine is the dose response is linear or not. While they see the trend of increasing effect with increasing arecoline concentration, the responses in Figure 4 during the dark period suggest that intermediate concentrations cause the greatest effect, not the highest concentration. Also, Figure A2 shows hyperactivity (above controls) with low arecoline concentrations while hypoactivity occurred at higher concentrations in dark conditions.
The authors understand the reviewer’s opinion about the possibility of a U-shaped dose-response of arecoline. After reconsidering several important points raised by the reviewer above, the authors agree that the determination of arecoline dose-response based on the current results is possible. Therefore, as the reviewer suggested, the possibility of non-linear (U-shaped) dose-response of arecoline’s explanation was added to several sections in the results part, which is section 2.1 and section 2.2. The explanation was added to these particular sections since the clearest evidence regarding the dose-response of arecoline is displayed in the results from these sections (LPMR and burst and total rotation counts), which are mentioned in detail by the reviewer above.
c. line 475-476 the authors state that they chose concentrations of arecoline so they would not cause ‘developmental retardation’. Please provide a reference for >1ppm causing developmental retardation. This information should also be moved earlier in the document, as it gives relevance to why the concentrations used were chosen.
Thank you for pointing out this matter. The authors realized that several references, which show arecoline in a concentration higher than 1 ppm causes developmental retardation, is needed to help the justification for choosing the concentrations used in the current study. Therefore, some previous findings that support this consideration had added to the manuscript, specifically on the second paragraph in the introduction part. One of these studies found that 100-400 ppm of arecoline caused general growth retardation and lower heart rate while another experiment showed that 10-400 ppm of arecoline generated developmental retardation and morphological deformities in zebrafish larvae.
Chang, B.E., et al., Developmental toxicity of arecoline, the major alkaloid in betel nuts, in zebrafish embryos. Birth Defects Research Part A: Clinical and Molecular Teratology, 2004. 70(1): p. 28-36.
Peng, W.-H., et al., Short-term exposure of zebrafish embryos to arecoline leads to retarded growth, motor impairment, and somite muscle fiber changes. Zebrafish, 2015. 12(1): p. 58-70.
d. line 524 mentions that arecoline can bind to dopaminergic receptors; however, this is not discussed further nor is there a reference given. How might this information affect the interpretation of your results?
The authors appreciated the reviewer for reminding this important issue. Actually, when the authors mentioned that arecoline can bind to dopaminergic receptors, it was based on a previous finding. However, the authors admit that the mistake was made so the reference was not added to the manuscript and thus, as the reviewer mentioned, it can affect the interpretation of your results. Therefore, to avoid this problem, the reference was added to that particular line in the discussion part.
Molinengo, L., M.C. Cassone, and M. Orsetti, Action of arecoline on the levels of acetylcholine, norepinephrine and dopamine in the mouse central nervous system. Pharmacology Biochemistry and Behavior, 1986. 24(6): p. 1801-1803.
e. paragraph between lines 510-536 discusses how different exposure times can cause different results. Was this observed in this study? Hyperactivity was seen after a 30min exposure and after a 24hr exposure, though it did vary depending on if activity was measured during light on or off.
Thank you for the question. In the present study, it was found that different exposure times cause different results. However, the word ‘different’ has a broad meaning. For example, ‘different’ means when a short exposure time causes hyperactivity and long exposure time leads to hyperactivity, such as ethanol exposure characteristic. Meanwhile, this word can also mean that even though both exposure times cause hyperactivity, it is still varied depending on if the activity was measured during light or dark cycles as the reviewer mentioned, which is observed in the present study. In addition, to enhance the quality of the discussion part, revision regarding this matter was conducted.
f. line 537-538 states “our result is consistent with the previous study showing the arecoline effect as a psychoactive and body excitability promotor”. What tests were performed to determine psychoactive effects? What is the difference between body excitability and hyperactivity?
The authors appreciated the questions. Based on a prior study, the term psychoactive effect means that the drugs, which is arecoline in this case, are capable to affect the behavior of an organism, whether it becomes hyperactivity or hypoactivity. Therefore, since hyperactivity behavior was observed in the treated fish by conducting the locomotor activity test, the authors believe that this kind of test is appropriate enough to conduct a psychoactive effect assay for drugs. Furthermore, regarding the following question, body excitement is the capability of an organism’s body into action as a state of excitement and it is also related to hyperactivity. However, this term is not only associated with locomotion, but also linked to the burst movement and average rotation counts measured in the present study. In addition, all of this information regarding the psychoactive effect and body excitability promotor of arecoline was added to the discussion part, specifically in the fifth paragraph.
Seifert, R. and B. Schirmer, A simple mechanistic terminology of psychoactive drugs: a proposal. Naunyn-Schmiedeberg's archives of pharmacology, 2020. 393(8): p. 1331-1339.
Williams, S., et al., Sociocultural aspects of areca nut use. Addiction biology, 2002. 7(1): p. 147-154.
17. What is ‘average rotation’ measuring? Is this turn angle or turning? How might this parameter be affected by the small size of the well in the 48 well plate?
Thank you for raising these important questions. In the present study, ZebraLab, the movement tracking software, counted the rotation done by the larvae. The rotation is based on the angle measured on the path of the animal and it was calculated throughout the test, counted by clockwise and counterclockwise, which in this case, they are combined together. The calculation of this endpoint was based on several parameters, which were minimum diameter and back angle. The minimum diameter is set to 5 mm, which was half of a single well’s diameter in a 48-well plate, and any rotation with a diameter greater than this value will be counted while all rotations below the given diameter will be disregarded. Meanwhile, regarding the back angle parameter, if the animal has started to rotate in a given way and turn back to rotate the other way, ZebraLab will keep the memory of the previous rotation angle until the animal has reached the value of the back angle. In case the back angle value is reached, which was set to 60° in the current study, the origin of the rotation will be reset. All of the values set on each parameter in this experiment were also advised by ViewPoint. Finally, this crucial information regarding the rotation count measurement was added to the materials and methods part, specifically in section 5.4.
http://www.viewpoint.fr/en/p/equipment/zebrabox-for-embryos-or-larvae/addons
18. Within the figures, it is hard to differentiate among the different symbols. The different colors help a lot, though.
The authors appreciate the suggestion. The authors understand that it is hard to differentiate the groups among the different symbols, especially in Figure 6 which is contains mAChR antagonist treatment results. Therefore, to solve this problem, the pattern in each data bar was removed, however, the authors replaced it with different colors since the reviewer suggested that different colors can help a lot to differentiate every group in the figure.
Reviewer 2 Report
This paper titled “Pharmaceutical Assessment Suggests Locomotion Hyperactivity in Zebrafish Triggered by Arecoline Might be Associated with Multiple Muscarinic Acetylcholine Receptors Activation” illustrates how a simple behavior in zebrafish larvae could be utilized in mechanistic drug studies. The authors have first shown that Arecoline induces hyperactivity in zebrafish larvae. Then, with the molecular docking method, they identified the hit compounds (by identifying the receptors) that could act as antagonists to Arecoline. By performing a rescue experiment with these antagonist compounds, the authors have tried to probe the mechanism of action of Arecoline.
Overall, the manuscript is good. The introduction has provided enough background to undertake this study. Methods are described well and the results are presented clearly. The authors have discussed their results taking into account the similarities and differences from other previous studies. The strong part of this manuscript is the utilization of the high-throughput behavioral methods integrated with computational model for the mechanistic drug study.
Some comments for the authors:
Line 12-13: This is unclear. Were the zebrafish exposed for 3 hours (acute exposre) at 96 hpf and experiment performed at 120 hpf?
Line123-128: Were the zebrafish larvae placed in fish water or drug solution during the experiment?
Line154-156: If the authors mention the reason(s) for selecting the 24 hour exposure method for the rescue effect, it would be of immense help to the readers.
Line237: Spell check “went”
Line 301-302: Do the authors think that short-term exposure is better than long-term exposure in the high throughput behavioral drug studies based on this data?
Is the movement seen in the Arecoline treated zebrafish larvae (Hyperactivity with increased burst movement and rotation) an epileptic movement?
Could the photo motor response induced by Arecoline related to the alteration of the action potential?
Spelling checks and editing throughout the manuscript will make this work better.
Author Response
This paper titled “Pharmaceutical Assessment Suggests Locomotion Hyperactivity in Zebrafish Triggered by Arecoline Might be Associated with Multiple Muscarinic Acetylcholine Receptors Activation” illustrates how a simple behavior in zebrafish larvae could be utilized in mechanistic drug studies. The authors have first shown that Arecoline induces hyperactivity in zebrafish larvae. Then, with the molecular docking method, they identified the hit compounds (by identifying the receptors) that could act as antagonists to Arecoline. By performing a rescue experiment with these antagonist compounds, the authors have tried to probe the mechanism of action of Arecoline.
Overall, the manuscript is good. The introduction has provided enough background to undertake this study. Methods are described well and the results are presented clearly. The authors have discussed their results taking into account the similarities and differences from other previous studies. The strong part of this manuscript is the utilization of the high-throughput behavioral methods integrated with computational model for the mechanistic drug study.
Some comments for the authors:
Line 12-13: This is unclear. Were the zebrafish exposed for 3 hours (acute exposre) at 96 hpf and experiment performed at 120 hpf?
Thank you for the questions. Actually, as the reviewer already knows, two exposure times of arecoline were applied in the present study, which was acute and chronic exposures. As the reviewer mentioned, it is true that for the chronic exposure, arecoline was administered at 96 hpf and the locomotor activity test was conducted at 120 hpf. However, for the acute exposure, the arecoline administration was carried out 30 minutes prior to the locomotor activity test which was conducted at 120 hpf. The whole paragraph regarding these exposure procedures in the materials and methods part was revised so it can provide better information.
Line123-128: Were the zebrafish larvae placed in fish water or drug solution during the experiment?
The authors appreciate the reviewer for asking this question. As it is already mentioned in the manuscript, zebrafish larvae were placed in a 9 cm-petri dish filled with the drug solution. Later, on 120 hpf (locomotor activity test), each larva was individually transferred to each well in a 48-well plate together with the drug solution from the petri dish. Therefore, the tested larvae were also exposed to the drug solution during the video recording in locomotion assay. This crucial information was added to the materials and methods part, specifically in section 5.1.
Line154-156: If the authors mention the reason(s) for selecting the 24 hour exposure method for the rescue effect, it would be of immense help to the readers.
Thank you very much for the suggestion. Actually, 24-hour exposure was chosen based on several prior studies that chronically exposed the zebrafish larvae to some chemicals, including arecoline. However, for the muscarinic acetylcholine receptors (mAChR) incubations, the authors found that there was a mistake regarding the usage of the word ‘rescue’. The authors believe that the more appropriate term for the current study’s methodology is ‘co-exposure’. Therefore, since the mAChR were co-exposed with arecoline, they were also exposed to the zebrafish larvae for the same time interval, which was 24 hours. The information regarding this matter was added to the materials and methods part, specifically in section 5.1.
Airhart, M. J., Lee, D. H., Wilson, T. D., Miller, B. E., Miller, M. N., & Skalko, R. G. (2007). Movement disorders and neurochemical changes in zebrafish larvae after bath exposure to fluoxetine (PROZAC). Neurotoxicology and teratology, 29(6), 652-664.
Peng, W.-H., et al., Short-term exposure of zebrafish embryos to arecoline leads to retarded growth, motor impairment, and somite muscle fiber changes. Zebrafish, 2015. 12(1): p. 58-70.
Line237: Spell check “went”
The authors appreciated the detailed correction. The authors admit that there was a mistyping in the mentioned line. The correct word should be ‘when’ instead of ‘went’, thus, the word was changed accordingly to avoid misunderstanding.
Line 301-302: Do the authors think that short-term exposure is better than long-term exposure in the high throughput behavioral drug studies based on this data?
Thank you for raising this question. Personally, the authors believe that word ‘better’ has a very broad meaning in behavior studies. Based on the results, the authors think that even though both acute and chronic exposures caused a hyperactivity behavior in zebrafish larvae, each exposure time has its own specific pattern of alteration that was dependent on if the activity was measured during light or dark cycles, thus, it is difficult to claim which time exposure is better in terms of the high-throughput behavioral drug studies. In addition, since the locomotor activity tests in the present study for both exposure time groups were conducted on 120 hpf of zebrafish larvae, there are no circumstances to state that one exposure time is better or less time-consuming than another.
Is the movement seen in the Arecoline treated zebrafish larvae (Hyperactivity with increased burst movement and rotation) an epileptic movement?
The authors appreciated the question. Based on a prior study in zebrafish, an epileptic movement is defined as spontaneous coils contractions associated with epileptic-like movement features, such as twitching, jitter, and tremor. Here, if the conclusion was drawn based on those mentioned behavioral endpoints (burst movement and rotation counts) only, there was a possibility that the alteration is included as an epileptic movement since these endpoints mainly focus on the activity and the angle measured on the path of the animal. However, a significantly high level of another behavioral endpoint, which was total distance traveled, observed in almost all of the arecoline-treated fish groups convince that the abnormality caused by arecoline is not an epileptic movement since this endpoint indicated that the treated fish exhibited a voluntary movement, which is the opposite of epileptic movement.
Basnet, R. M., Guarienti, M., & Memo, M. (2017). Zebrafish embryo as an in vivo model for behavioral and pharmacological characterization of methylxanthine drugs. International journal of molecular sciences, 18(3), 596.
Could the photo motor response induced by Arecoline related to the alteration of the action potential?
Thank you for the question. To the best of the authors’ knowledge, an action potential is defined as a sudden, fast, transitory, and propagating change of the resting membrane potential and it occurs when the membrane potential of a specific cell location rapidly rises and falls. This process occurs on neurons and muscle cells by sending a signal that can connect with other neurons. Regarding the question from the reviewer, the abnormalities in zebrafish larval photo motor response induced by arecoline might be related to the alteration of the action potential since this response was closely related to the neurons and muscle cells. Therefore, it is intriguing to study this hypothesis in future studies. In addition, this possibility of the relation between photo motor response induced by arecoline and alteration of the action potential was included in the discussion part, specifically in the second paragraph.
Hodgkin, A. L., & Huxley, A. F. (1952). A quantitative description of membrane current and its application to conduction and excitation in nerve. The Journal of physiology, 117(4), 500-544.
Spelling checks and editing throughout the manuscript will make this work better.
The authors appreciated the input. Therefore, the authors had tried their best to check the spelling and edit throughout the manuscript to make it better as the reviewer suggested. The authors hope that the updated version of the manuscript is significantly improved than the previous one.
This manuscript is a resubmission of an earlier submission. The following is a list of the peer review reports and author responses from that submission.